# Critic Identifiability in Offline Reinforcement Learning with a Deterministic Exploration Policy

## Abstract

Offline Reinforcement Learning (RL) promises to enable the adoption of RL in settings where logged interaction data is abundant but running live experiments is costly or impossible. The setting where data was gathered with a stochastic exploration policy has been extensively studied, however; in practice, log data is often generated by a deterministic policy. In this work, we examine this deterministic offline RL setting from both a theoretical and practical perspective. We describe the critic identifiability problem from a theoretical standpoint, arguing that algorithms designed for stochastic exploration are ostensibly unsuited for the deterministic version of the problem. We elucidate the problem further using a set of experiments on contextual bandits as well as continuous control problems. We conclude that, quite surprisingly, the tools for stochastic offline RL, notably the TD3+BC algorithm, are applicable after all.

## 1 Introduction

In many practical settings where we want to apply Reinforcement Learning, running live experiments is costly. For example, in recommender systems, running an experiment with a new policy of unknown quality might lead to a poor user experience and risks losing revenue. Even more starkly, in healthcare, ethical considerations may completely preclude executing policies with unknown performance. Offline Reinforcement Learning promises to address this problem by enabling us to learn only from logged data that we already have, without having to query additional interactions from the environment.

Offline RL algorithms (Fujimoto et al., 2019; Fujimoto & Gu, 2021; Kostrikov et al., 2021; Kumar et al., 2020; Fu et al., 2022) often work by learning a critic and an actor network, with the critic attempting to estimate the quality of the actor's policy and the actor attempting to improve the policy using values learned by the critic. Under regularity conditions, this process is known to lead to policies with improved returns (Sutton et al., 1999; Silver et al., 2014). Crucially, in the variants of these methods most commonly used in practice, the critic depends on both a state and an action. When exploration data comes from a stochastic policy, there is ample data to train the critic since the sampled actions span the whole action space.

However, wherever the exploration policy is deterministic, by definition we only have one action per state to learn the critic. This means that we have to depend on non-trivial generalization properties of the critic network to obtain useful estimates of the policy gradient. In this paper, we conduct an extensive study of this problem, with the aim of producing an offline Reinforcement Learning agent with good performance on such deterministic data.

**Contributions**   We make progress on deterministic offline RL in two main ways. First, we describe the critic identifiability problem from a theoretical standpoint. Second, using a set of continuous control benchmarks, we show empirically how the problem can be addressed by weight initialization and the dynamics of neural network optimization alone. Lastly, we conclude by recommending TD3+BC-Phased, a variation of the TD3+BC algorithm (Fujimoto & Gu, 2021) for performing offline RL from data generated by deterministic policies. To aid reproducibility, all our MuJoCo experiments conform to the recommendations of Agarwal et al. (2021).

## 2 Background

**Markov Decision Process**  We formulate the RL problem using a Markov Decision Process (MDP). An MDP is a tuple $(\mathcal{S}, \mathcal{A}, r, P, \gamma, \mathcal{I})$ where $\mathcal{S}$ is the state space, $\mathcal{A}$ is the action space, $r$ is a reward function, $P$ is the MDP's transition dynamics probability distribution, $\gamma$ is a discount factor parameter and $\mathcal{I}$ is the distribution over initial states (Puterman, 2014). A policy $\pi$ is a mapping from states to distributions over actions, with $\int_{a \in \mathcal{A}} \pi(a|s) da = 1$, $\forall s \in \mathcal{S}$. For deterministic policies, we abuse notation by treating $\pi$ as a function $\mathcal{S} \to \mathcal{A}$.

**Dataset**  We study the problem of offline RL (Levine et al., 2020) for continuous state and action spaces. In offline RL, a fixed dataset of data previously gathered in $N$ episodes of length $T$, $D_N = \{(s_t^i, a_t^i, r_{t+1}^i, s_{t+1}^i) : t = 1, \ldots, T, i = 1, \ldots, N\}$ is used for learning. No new data is gathered at the time of learning[1]. This is in contrast to online RL in which the agent interacts with an environment, gathers data, and learns from the data while it is being gathered.

**Value Functions and the Return**  We define the return to be the sum of the future discounted rewards. The value of a state $s$ under a policy $\pi$ is defined as the expected return, given that the agent starts in state $s$ and follows $\pi$ thereafter:

$$V^\pi(s) \stackrel{\text{def}}{=} \mathbb{E}_\pi \left[ \sum_{t=0}^T \gamma^t r(S_t, A_t) | S_0 = s \right].$$

Similarly, the state-action value function is defined as expected return if an agent starts from state $s$, takes action $a$, and follows policy $\pi$ thereafter:

$$Q^\pi(s, a) \stackrel{\text{def}}{=} \mathbb{E}_\pi \left[ \sum_{t=0}^T \gamma^t r(S_t, A_t) | S_0 = s, A_0 = a \right].$$

We skip the subscripts denoting the policy in cases it is clear what policy is meant.

**Actor-Critic Algorithms**  Our focus is on the policy gradient family of algorithms (Sutton et al., 1999), in which the policy is parameterized. We write $\pi_{\boldsymbol{\theta}}(s)$ to denote a deterministic policy parameterized by a neural network whose parameters are denoted by $\boldsymbol{\theta}$. The neural network that estimates probabilities of actions is often referred to as the *actor* network. In policy gradient algorithms, the goal is to learn the policy parameters by doing gradient ascent on the following objective function:

$$J(\boldsymbol{\theta}) \stackrel{\text{def}}{=} \mathbb{E}_{s_0 \sim \mathcal{I}}[V^{\pi_{\boldsymbol{\theta}}}(s_0)],$$

where $\mathcal{I}$ is the distribution from which the initial state is sampled. The gradient of this objective is then used to improve a performance measure. Silver et al. (2014) show in their *deterministic policy gradient theorem* that:

$$\nabla_{\boldsymbol{\theta}} J(\boldsymbol{\theta}) = \mathbb{E}_{s \sim \mu_{\pi_{\boldsymbol{\theta}}}} [\nabla_a Q^{\pi_{\boldsymbol{\theta}}}(s, a)|_{a = \pi_{\boldsymbol{\theta}}(s)} \nabla_{\boldsymbol{\theta}} \pi_{\boldsymbol{\theta}}(s)], \tag{1}$$

where $\mu_{\pi_{\boldsymbol{\theta}}}$ is the discounted occupancy measure induced by the deterministic policy $\pi_{\boldsymbol{\theta}}(s)$ and defined as $\mu_\pi(s') = \int_{\mathcal{S}} \sum_{t=1}^T \gamma^{t-1} \mathcal{I}(s) p(s \to s', t, \pi) ds$, where $p(s \to s', t, \pi)$ is the measure associated with being in state $s'$ after $t$ transitions starting in state $s$ and following policy $\pi$.

In the actor-critic family of algorithms, an estimate of state-action value function is also learned that directs the updates of policy parameters. The state-action value function is typically parameterized using a second neural network, called the *critic* network, $\hat{Q}_{\boldsymbol{\phi}}$, whose parameters are denoted by $\boldsymbol{\phi}$. Algorithms like DDPG (Lillicrap et al., 2015) and TD3 (Fujimoto et al., 2018), work by making two approximations in (1). First, they replace $Q_{\boldsymbol{\theta}}$ by $\hat{Q}_{\boldsymbol{\phi}}$. Second, they replace $\mu_{\pi_{\boldsymbol{\theta}}}$ with $p_{\pi_{\boldsymbol{\theta}}}$, the long-term distribution of being in a state defined as $p_\pi(s') = \int_{\mathcal{S}} \sum_{t=1}^T \mathcal{I}(s) p(s \to s', t, \pi) ds$, which does not factor in the discount. This leads to the approximate update for the policy gradient

$$\nabla_{\boldsymbol{\theta}} J(\boldsymbol{\theta}) \approx \mathbb{E}_{s \sim p_{\pi_{\boldsymbol{\theta}}}} \left[ \nabla_a \hat{Q}_{\boldsymbol{\phi}}(s, a)|_{a = \pi_{\boldsymbol{\theta}}(s)} \nabla_{\boldsymbol{\theta}} \pi_{\boldsymbol{\theta}}(s) \right]. \tag{2}$$

---

[1]Our notation assumes episode length $T$ to be the same for all episodes for ease of exposition. The extension to episodes of varying lengths is straightforward.

This is often written it in an equivalent form as $\mathbb{E}_{s \sim p_{\pi_{\boldsymbol{\theta}}}}[\nabla_{\boldsymbol{\theta}} \hat{Q}_{\boldsymbol{\phi}}(s, \pi_{\boldsymbol{\theta}}(s))]$, which is equal to the right hand side of (2). In batch RL algorithms, the state distribution is (heuristically) replaced by the dataset $D$, giving the update $\mathbb{E}_{s \sim D}[\nabla_{\boldsymbol{\theta}} \hat{Q}_{\boldsymbol{\phi}}(s, \pi_{\boldsymbol{\theta}}(s))]$.

**TD3+BC Actor Update**   The TD3+BC algorithm (Fujimoto & Gu, 2021) adds a behavior cloning term to the TD3 update:

$$\max_{\boldsymbol{\theta}} \mathbb{E}_{(s,a) \sim D} \left[ \lambda \hat{Q}_{\boldsymbol{\phi}}(s, \pi_{\boldsymbol{\theta}}(s)) - \left\| \pi_{\boldsymbol{\theta}}(s) - a \right\|_2^2 \right], \tag{3}$$

where the action $a$ could be multi-dimensional and $\lambda$ is defined below. Equation 3 can be justified as follows. We want to use policy gradients as per equation 2. However, to minimize learning bias in the deep RL setting, it is important that the policy we learn stays close to the data, so we can be sure about its performance. Ideally, we want to encode this by adding a hard constraint saying that a divergence between the learned policy and the data-generating policy is smaller than $\epsilon$ (Schulman et al., 2015; Laroche et al., 2019; Wu et al., 2019; Jaques et al., 2019; Kumar et al., 2019). For stochastic policies, we could use the KL divergence between Gaussian distributions with the same spherical covariance, and for deterministic policies, we could use the squared $L_2$-Wasserstein divergence. In either case, we just get the mean squared error back (up to a multiplicative constant), which brings us back to (3). Instead of using the theory of optimization to find the Lagrange multiplier $\lambda$, Fujimoto & Gu (2021) heuristically propose to set $\lambda$ using a schedule that has been shown to work remarkably well in practice:

$$\lambda \leftarrow \frac{\alpha}{\frac{1}{|\mathcal{B}|} \sum_{(s_i, a_i) \in \mathcal{B}} |\hat{Q}_{\boldsymbol{\phi}}(s_i, a_i)|}, \tag{4}$$

where $\mathcal{B}$ is the mini-batch being used at the current time step, $|\mathcal{B}|$ is the number of transitions in the mini-batch, and $\alpha$ is a tuneable hyperparameter. In addition, in order to improve the stability of the algorithm, a target actor $\pi^{\text{target}}$ is also learned in parallel. This target policy is defined by its weights $\boldsymbol{\theta}^{\text{target}}$, updated as $\boldsymbol{\theta}^{\text{target}} \leftarrow \tau \boldsymbol{\theta} + (1 - \tau) \boldsymbol{\theta}^{\text{target}}$.

**TD3+BC Critic Update**   Before closing the Background section, we provide the update rules for the critic update used by TD3 and TD3+BC algorithms. Both algorithms use an update inspired by double Q-learning (Hasselt, 2010) to update their critic. This reduces the overestimation bias incurred by the original Q-learning algorithm. In addition, for stability, the two critic functions have two sets of parameters each: $\boldsymbol{\phi}_i$, which is being learned directly and $\boldsymbol{\phi}_i^{\text{target}}$, which lags the learned parameters and is updated as $\boldsymbol{\phi}_i^{\text{target}} \leftarrow \tau \boldsymbol{\phi}_i + (1 - \tau) \boldsymbol{\phi}_i^{\text{target}}$. Two sets of parameters, $\boldsymbol{\phi}_1$ and $\boldsymbol{\phi}_2$ are learned; both functions are trained to match

$$y(r, s', a') \leftarrow r + \min_{i=1,2} \hat{Q}_{\boldsymbol{\phi}_i^{\text{target}}}(s', a')$$

where both parameter sets are learned using regression, by minimizing the following objective function:

$$L(\boldsymbol{\phi}_i, D) \stackrel{\text{def}}{=} \mathbb{E}_{(s,a,r,s') \sim D, a' \sim \pi^{\text{target}}(s')} \left[ \left( \hat{Q}_{\boldsymbol{\phi}_i}(s, a) - y(r, s', a') \right)^2 \right]. \tag{5}$$

Finally, Fujimoto & Gu (2021) always use $\hat{Q}_{\boldsymbol{\phi}_1}$ in the policy learning step. Learning $\hat{Q}_{\boldsymbol{\phi}_2}$ has no use beyond helping to reduce the overestimation bias.

## 3   Related Work

As far as we are aware, ours is the first work that studies batch Reinforcement Learning in a setting where the exploration policy is deterministic. The closest related work is literature on batch RL algorithms. The most significant batch algorithm from the perspective of our work is TD3+BC (Fujimoto & Gu, 2021), which was already described in the Background section. For a more complete recent overview of batch RL with stochastic policies, we refer the reader to the work by Fu et al. (2022).

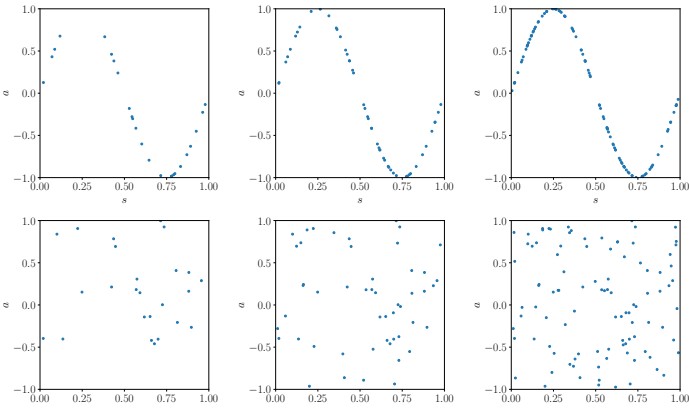

Figure 1: Available state-action space data on a contextual bandit problem with 25, 50, 100 interactions for deterministic exploration (top row) and stochastic exploration (bottom row).

**Batch RL with Finite Datasets**  Fujimoto et al. (2019) extensively study the setting where the dataset given to a batch RL agent is finite, analyzing the difference between an approximation to the optimal policy computed on the finite dataset and the true optimal policy. On the surface it seems that this work should be similar to our setting of deterministic exploration since, for continuous state-action spaces, a stochastic policy will see only one action in a given state. However, in reality, our setting is very different. A major distinguishing feature is that, for a deterministic exploration policy, we may fail to explore the whole state-action space even given we visit every state and are provided with an infinite number for trajectories. Therefore, the tools used to study finite datasets generated by stochastic policies will not be enough to capture all phenomena characteristic of deterministic policies. To explain this point further, in Figure 1, we show the difference in the available data on a contextual bandit problem between the setting of a stochastic and deterministic exploration policy.

**Concentrability and Theoretical Batch RL Work**  Munos & Szepesvári (2008) use a concentrability coefficient to study Fitted Value Iteration. Chen & Jiang (2019); Xie & Jiang (2020); Rashidinejad et al. (2021) later use the same idea to facilitate the analysis of batch RL in the context of a finite dataset. Crucially, in all of these works, the concentrability coefficient divides by the probability of exploring a given state-action pair, thus implicitly assuming that the policy that generated the data is stochastic. Similarly, Yin et al. (2021) use the minimum probability of seeing a state-action pair in exploration to study off-policy evaluation, while Yin & Wang (2021) also use it to study model-based batch RL, again constraining both approaches to stochastic exploration policies. On the other hand, Xie et al. (2021) and Cheng et al. (2022) use a characterization of distribution shift which is aware of the function class used to approximate the value functions, allowing for deterministic exploration policies. While this work is highly relevant, it follows somewhat different goals than we do. Xie et al. (2021) provide a purely theoretical analysis, without testing the ideas in practice. On the other hand, Cheng et al. (2022) are interested in deriving a novel batch RL algorithm that leverages a notion of pessimism and can be theoretically argued to work under a broad spectrum of circumstances. Our goals are different: the main questions we ask are: (1) empirically, can we use off-the-shelf tools that work well with stochastic exploration policies to solve problems where data is generated by deterministic policies and (2) theoretically, under what assumptions can we learn an accurate model of the Q function given a deterministic exploration policy.

**Phased and One-Step Algorithms**  The idea of separating out critic and actor learning into separate phases, deployed until approximate convergence, as opposed to learning the actor and the critic simultaneously, has been studied by Peng et al. (2019). The concept of doing one loop of policy iteration rather than many in the context of batch Reinforcement Learning has been studied by Brandfonbrener et al. (2021). They identify that, the reason why such 'single iteration' algorithms can perform better than their 'multi-iteration' equivalents is that they do not require off-policy evaluation, which is extremely unstable and prone

to inaccuracy in the deep RL setting. Our TD3+BC-Phased algorithm is a special case of the algorithm used by Brandfonbrener et al. (2021) and we do not claim algorithmic novelty, only that the learning from algorithms deployed on data generated by stochastic exploration policies translate to deterministic exploration as well (we believe our paper to be the first one that experimentally studies deep batch RL in the context of deterministic exploration).

**Additional Regularizers**  A powerful idea in the batch RL literature is to add additional regularizers to an online RL algorithm. For example, CQL (Kumar et al., 2020) adds an additional term to the critic loss, so that the critic is learning an (approximate) lower bound on the true $Q$ function to both prevent overestimation and constrain the policy to stay near the training data. An extension to this idea is presented by COMBO (Yu et al., 2021), which uses a model to construct a more accurate lower bound. While our analysis in section 7 is also based on additional regularization, we address a different problem. Indeed, none of these prior works addresses the question of what happens if the exploration policy is deterministic, which is the main focus of our paper.

**Critic Generalization**  Implicit Q-Learning (Kostrikov et al., 2021), like our algorithm, heavily relies on generalization properties of the function approximator used for the critic. It also separates out the stages of learning the critic and actor. However, Implicit Q-learning has been designed with stochastic exploration policies in mind, and, unlike our work, does not consider the problem that the critic might not be identifiable under deterministic exploration.

**Imitation Learning**  Imitation Learning algorithms ignore the reward signal completely, learning a policy by performing supervised learning of the demonstration data. In sequential decision-making problems, due to error compounding, imitation learners suffer from poor performance (Ross & Bagnell, 2010). While Behavioral Cloning can work well for policies generated by deterministic experts, it is constrained by the availability of high-quality data. Recently, a new baseline has been proposed (Chen et al., 2021) that uses a reward signal. Percent Behavioral Cloning performs supervised learning on a subset of data with sufficiently high returns. However, unlike the batch RL setting which we study, this technique is still fundamentally limited by the fact that the behavior policy has to solve the same task learned by the agent.

## 4  Critic Identifiability

In this section, we will describe the problem of critic identifiability for MDPs. We analyze model-free offline RL algorithms which learn a critic. In continuous control problems, the critic usually attempts to approximate the $Q$-function of a policy, which, crucially, depends both on a state and an action. In offline RL, data typically comes from a stochastic policy with support on the whole action space. In this case, given adequate state coverage and a smoothness assumption on the $Q$-function, the critic is identifiable in the sense that, as the amount of training data approaches infinity, the learned critic becomes close to the true $Q$-function. We can formalize this notion using the following definition, where we denote the set of datasets with $\mathcal{D}$ and the function class used to learn the critic with $\mathcal{F}$.

**Definition 1** (Identifiability)**.** Consider a problem class (i.e. a set of MDPs) $\mathcal{P}$. The problem class is *identifiable* in region $\mathcal{R} \subset \mathcal{S} \times \mathcal{A}$ with exploration policy class $\Pi$ and learning algorithm $\text{Alg} : \mathcal{D} \to \mathcal{F}$ if for every possible MDP $M \in \mathcal{P}$, for every $\epsilon > 0$ and $\delta \in (0, 1]$ there exists an exploration policy $\pi_{\text{E}}^{\epsilon,\delta} \in \Pi$ and a dataset size $n^\epsilon$ so that for all $N \geq n^\epsilon$ with probability at least $1 - \delta$ we have

$$\sup_{(s,a)\in\mathcal{R}} |Q_M^{\pi_{\text{E}}^{\epsilon,\delta}}(s, a) - \hat{Q}_N^{\epsilon,\delta}(s, a)| \leq \epsilon.$$

Here, $\hat{Q}_N^{\epsilon,\delta} = \text{Alg}(D_N^{\pi_{\text{E}}^{\epsilon,\delta}})$ denotes a critic trained using algorithm Alg on a dataset $D_N^{\pi_{\text{E}}^{\epsilon,\delta}}$ of $N$ episodes gathered with policy $\pi_{\text{E}}^{\epsilon,\delta}$. $Q_M^{\pi_{\text{E}}^{\epsilon,\delta}}$ denotes the ground truth Q-value of the policy $\pi_{\text{E}}^{\epsilon,\delta}$ in the MDP $M$.

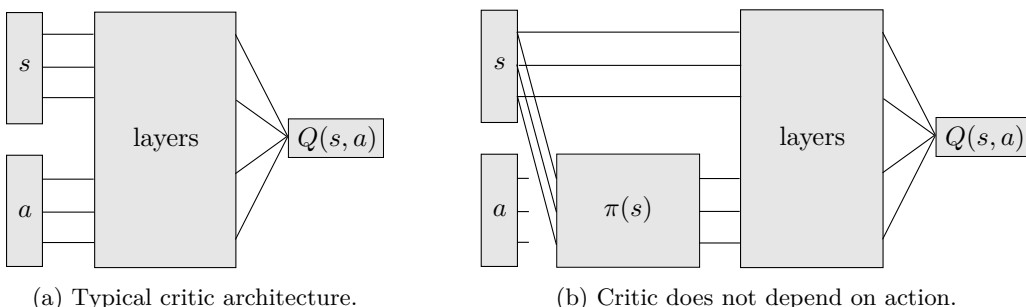

(a) Typical critic architecture.  (b) Critic does not depend on action.

Figure 2: An illustration of the critic identifiability problem.

In other words, informally, a problem class is identifiable with a given exploration policy class and a learning algorithm if we can accurately learn the Q-function of an exploration policy in the class using the algorithm[2].

For ease of exposition, we will now focus on contextual bandits, where the ground truth $Q$ function and the reward function are the same and the episode length is one. First, consider the case where the exploration policy is stochastic (with support on the whole action space), where we make a Lipschitz smoothness assumption on the critic (we denote the set of Lipschitz functions that fulfill certain regularity conditions with $\mathcal{H}$ — see Appendix B for the exact definition), and use a learning algorithm that returns any Lipschitz-smooth function minimizing MSE error on the training set

$$\text{Alg}(D_N) = \underset{\hat{Q} \in \mathcal{H}}{\arg\min} \frac{1}{N} \sum_{(s,a,r) \in D_N} (\hat{Q}(s,a) - r)^2. \tag{6}$$

Moreover, we assume that the problem is realizable, which implies that the term being minimized is always zero. In this case, identifiability on $\mathcal{S} \times \mathcal{A}$ follows directly from standard Rademacher bounds (von Luxburg & Bousquet, 2004) in the sense that we have $\lim_{N \to \infty} \sup_{(s,a) \in \mathcal{S} \times \mathcal{A}} |Q(s,a) - \hat{Q}_N(s,a)| = 0$ with probability one for every such non-degenerate exploration policy. While other regularity assumptions may lead to faster convergence of this limit, in this work we limit ourselves to the Lipschitz assumption due to the fact that is is relatively straightforward to ensure in the context of deep learning.

However, identifiability is no longer a given when the exploration policy is deterministic. To see an illustration why, recall that the critic typically has an architecture similar to the one illustrated in Figure 2a. Now, for deterministic policies, the action can be completely determined from the state. As shown in Figure 2b, this means that a critic trained using a MSE loss can simply learn a copy of the policy and use it whenever the $Q$ function depends on the action. Crucially, if this happens, the action input to the network is completely ignored. This means that policy gradients obtained using that critic are useless, since the gradient of the critic with respect to the action input is zero. More broadly, since the MSE loss function is computed on the training set where we only have one action per state, the learned critic will, without further assumptions, achieve arbitrary values for actions other than the one taken by the exploration policy.

The central finding of this work is that, quite surprisingly, **the inductive bias arising from using critics represented by ReLU networks and learned using the Adam algorithm (Kingma & Ba, 2014) is enough to provide adequate generalization and useful policy gradients**. In other words, despite the fact that the MSE loss does not distinguish a critic network that provides useful policy gradients from one that exhibits the pathology shown in Figure 2b, the optimizer always converges to a network with useful policy gradients. We examine this phenomenon empirically in the next section.

---

[2]We will explain in Section 6 that for a nontrivial class of problems, it is sufficient in practice to learn the Q function of the exploration policy and we do not have to learn the Q function of an arbitrary policy.

# 5 Experiments on a Contextual Bandit

## 5.1 Generalization and Critic Identifiability

In order to provide a fuller picture of the critic identifiability problem, we studied it empirically for contextual bandits. Figure 3a shows three instances of a contextual bandit, with different reward functions (which are also the ground truth $Q$ functions). Specifically, we studied the quadratic, ring and 'three pits' reward functions. The quadratic function is the most representative since all smooth functions are approximately locally quadratic, while the other two, more complex, examples can be used to study the limits of generalization of non-local features. We generated a training set using a policy tracing a re-scaled sine wave, visible in the figure as a black curve, and obtained the corresponding actions by evaluating it on an evenly distributed grid of 100 states.

We then used this training set to learn the critics shown in Figure 3b, using a ReLU network and the Adam optimizer (see appendix A for the learning learning rate and other technical details). Surprisingly, for the quadratic function, despite the training set being confined to the policy, the generalization ability of the ReLU network combined with the optimizer turned out to be outstanding, recovering the shape of the function even in points far away from the policy that generated the data. We repeated this experiment multiple times and the results were always qualitatively the same. Each time, we obtained good generalization, which implies good-quality policy gradients in the neighborhood of the exploration policy. On the other hand, the conclusion from plots for the ring reward and the 'four pits' reward is that we only get a reasonable approximation to the true Q in regions of the state-action space close to the policy that generated the data and varying results elsewhere. This is to be expected: while the inductive bias inherent in ReLU networks trained with Adam can be expected to encode a notion of continuity, it cannot make up features of the reward landscape completely absent from the training set. This can be witnessed in the plot showing the part of Figure 3b showing the 'four pits' reward function, where the critic was only able to learn about two of the four pits, since the other two were not represented in the training data.

While Figure 3 provides a qualitative measure of generalization, we also provide a quantitative one in Table 1. It can be seen that for all reward functions, we are able to generalize very well when evaluated on state-action pairs coming from the exploration policy and acceptably well for state-action pairs that come from near the policy. On the other hand, generalization to regions of state-action space very far form the exploration policy is only sometimes possible. We provide details of the experiment used to generate Table 1 in Appendix A.

Overall, since we found the high quality of the generalization perplexing (especially for the quadratic critic), we wanted to further investigate whether the critic identifiability problem can become an issue in practice. To shed light on this, we ran another experiment. We learned another critic function, keeping the same network architecture and optimizer settings, but forcing the weights multiplying the action to be zero at every optimization step. Figure 3c shows the result. By design, the obtained critic function does not depend on the action and produces completely useless policy gradients. However, it achieves the same loss of 0.001 on the training set as the network shown in Figure 3b. This confirms in practice the problem identified in the previous section: it is possible to minimize the MSE loss well while still having a critic useless for policy updates.

Given the current limited understanding of the generalization of neural networks, it is hard to speculate why exactly the combined effect of the network architecture, weight initialization and the optimizer always end up preferring weights giving rise to the good behavior in 3b above the pathological one shown in 3c. However, if we treat the number of required optimization steps required to learn the critic as a measure of simplicity of the learned function, it turns out that the good critics are indeed simpler, requiring on the order of 30%-50% optimization steps to train.

## 5.2 The Critic Identifiability Problem Occurs in Practice

Above, we could see that the critic identifiability problem can arise in the sense that a good learned critic and the pathological one have the same MSE error on the training set. However, in order to obtain the

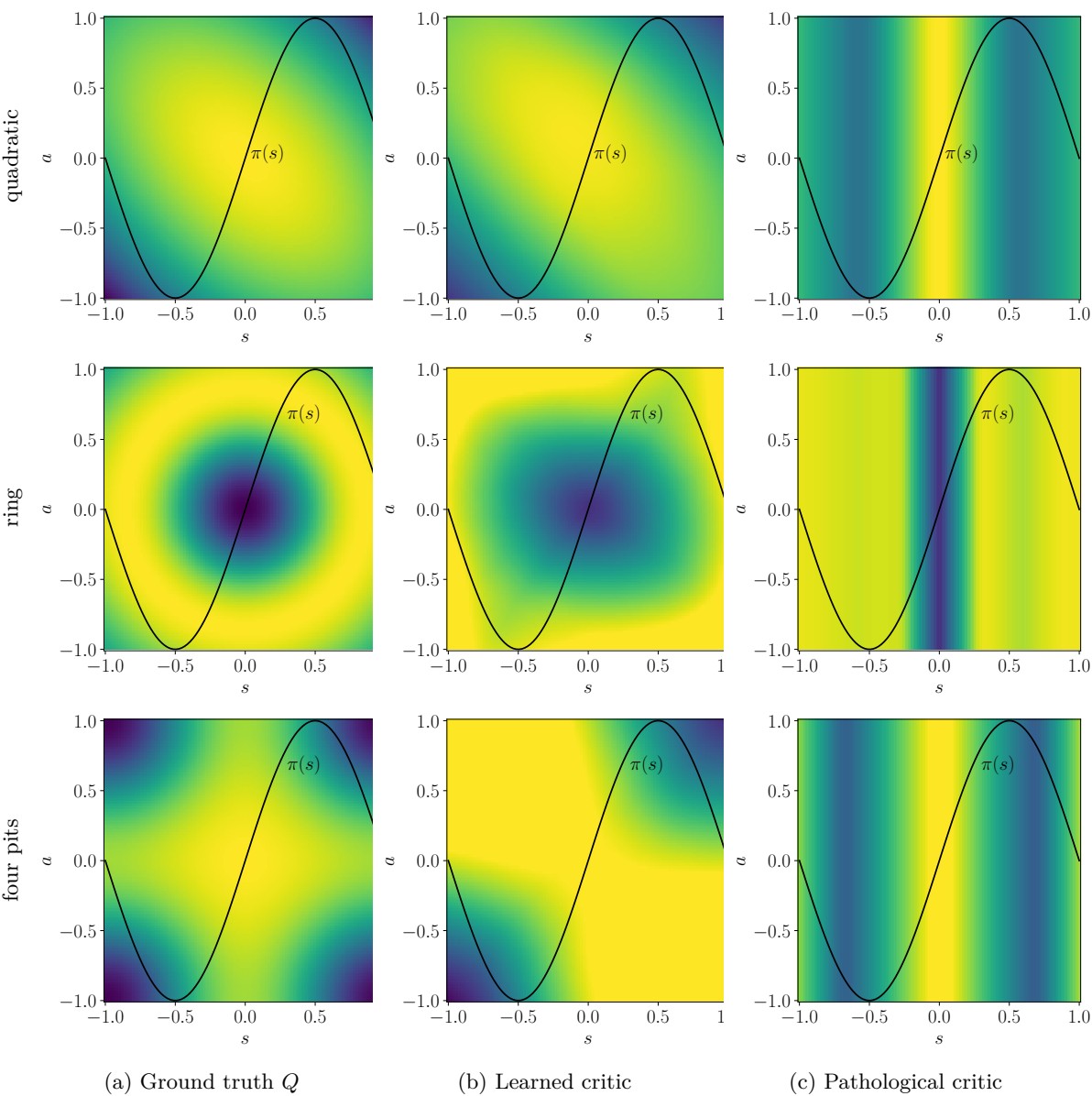

Figure 3: Experiments with critic identifiability. Training data comes only from the policy $\pi(s)$.

| Reward Type | Matching $\pi_{\mathbf{E}}$ | Near $\pi_{\mathbf{E}}$ | Far From $\pi_{\mathbf{E}}$ |
|:---:|:---:|:---:|:---:|
| **quadratic** | $0.0010 \pm 0.0000$ | $0.0015 \pm 0.0000$ | $0.0081 \pm 0.0004$ |
| **ring** | $0.0008 \pm 0.0000$ | $0.0012 \pm 0.0000$ | $0.0062 \pm 0.0002$ |
| **four pits** | $0.0009 \pm 0.0000$ | $0.0010 \pm 0.0000$ | $0.1564 \pm 0.0067$ |

Table 1: Mean squared generalization error of critic networks trained using data coming from a deterministic policy $\pi_{\mathbf{E}}$ on various reward signals.

pathological critic, we had to artificially constrain the critic network not to depend on the action, which can be argued to be unnatural. In this section, we show that, with linear function approximation, the critic identifiability problem can organically happen, even for linear critics.

Indeed, consider the class of contextual bandits, with states in $\mathbb{R}^{10}$ and actions in $\mathbb{R}^9$, where the dataset is generated by the deterministic policy

$$\pi_{\mathrm{E}}([s_1, s_2, \ldots, s_{10}]^\top) = [s_1, s_2, \ldots, s_9]^\top$$

and the ground truth Q (the reward) is defined as

$$Q([s_1, s_2, \ldots, s_{10}]^\top, [a_1, a_2, \ldots, a_9]^\top) = \sum_{i=1}^{9} a_i + s_{10}. \tag{7}$$

Intuitively, this setting encapsulates the problem commonly found when learning a critic function in a high-dimensional state action space. The dataset lies on a low-dimensional manifold and there are many candidate critics in the (linear in this case) function class that model the training data with zero error. However, most of them will generalize very poorly outside of this training dataset. Indeed, we experimentally verified that, for this example, gradient descent trained on data generated by $\pi_{\mathrm{E}}$ not only computed a completely wrong approximation to the ground truth Q, but gave policy gradients where at least one gradient coordinate points in direction opposite to the correct one (details of the experiment are in Appendix A).

## 6  TD3+BC-Phased Algorithm

While the discussion in the previous section suggests that we could use unmodified TD3+BC, it turns out that the setting of deterministic exploration policy allows us to make one further simplification to the algorithm.

Similarly to most other actor-critic methods, TD3+BC learns the critic and the actor simultaneously. Specifically, the critic learns the Q-function of the actor policy and the actor learns a policy that maximizes the current critic estimate. Together, this process represents an incremental version of policy iteration. However, if the exploration policy is deterministic, it is reasonable to instead confine ourselves to a single step of policy iteration, where the critic learns the value of the exploration policy and the actor improves on just this value. This simplification is possible because, for deterministic policies, we by definition do not have the data about values of actions other than the one chosen by the exploration policy, making the evaluation of other policies tricky. On the other hand, of course one could also argue that we could instead rely on critic generalization to facilitate multiple steps of policy improvement.

Ultimately, given the current understanding of actor-critic algorithms, the question of whether it makes sense to perform more than one step of policy iteration can only be answered empirically. We did just that, by comparing two algorithms: regular TD3+BC and our modification, which we call TD3+BC-Phased. TD3+BC-Phased is described in Algorithm 1 and proceeds in three stages. First, the exploration policy is distilled from data using behavioral cloning. Second, a critic is learned to evaluate this policy.

$$L(\phi_i, D) \stackrel{\text{def}}{=} \mathbb{E}_{(s,a,r,s') \sim D} \left[ \left( Q_{\phi_i}(s, a) - y(r, s', \hat{\pi}_E(s')) \right)^2 \right]. \tag{8}$$

This update is different from equation equation 5 in that the policy whose value the critic is computing is an approximation $\hat{\pi}_E$ to the policy that generated the data as opposed to an approximation to the optimal policy the algorithm is learning about. Finally, an actor network is trained to maximize the value of the critic. This staged algorithm is simpler than vanilla TD3+BC because the critic does not depend on the actor. While this algorithm is a special case of the algorithm framework introduced by Brandfonbrener et al. (2021) in that we only do one iteration of policy improvement, our experiment is different in that we consider data generated by a deterministic exploration policy, while Brandfonbrener et al. (2021) uses stochastic exploration policies.

In Figure 4, we report the performance of both variants of the algorithm on a variant of the D4RL benchmark where the demonstration data was generated by deterministic policies (see Appendix A for details), to match the setting in our paper. It can be seen that the phased version of the algorithm has better performance, on both expert and medium datasets. We therefore chose this version as a basis for our later experiments with regularization, described in Section 8.

---

**Algorithm 1** The TD3+BC-Phased Algorithm

---

Learn $\hat{\pi}_E$ by minimizing $\sum_{(s,a)\in D_n} \|a - \pi_\theta(s)\|$ wrt. $\theta$
Learn $\hat{Q}$ by minimizing equation 8
Learn $\pi$ by minimizing equation 3

---

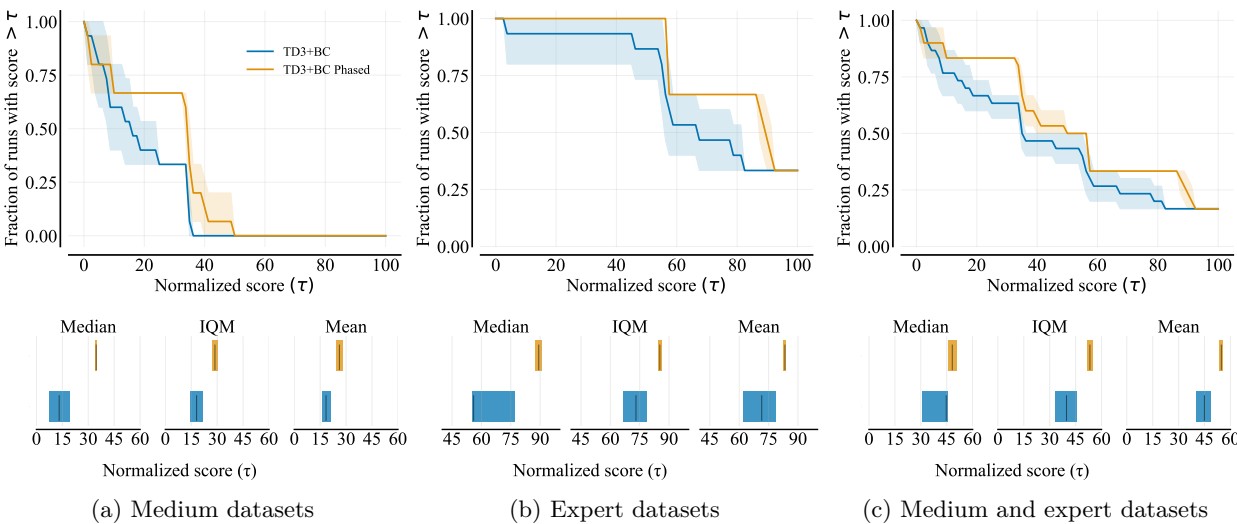

(a) Medium datasets      (b) Expert datasets      (c) Medium and expert datasets

Figure 4: Comparison of vanilla TD3+BC (in blue) and the phased version (in orange). Performance profiles and aggregate metrics (mean, interquartile mean, median) were computed using 5 seeds in each of 3 environments (MuJoCo Walker, Hopper and HalfCheetah). 95%-confidence interval were obtained via stratified bootstrap with 50k replications.

## 7 Ensuring Critic Identifiability with Lipschitz Regularization

In order to mitigate the critic identifiability problem, we propose to constrain the critic $\hat{Q}$ to be Lipschitz-continuous with constant $L$. We propose to use Lipschitz continuity as a regularity condition that allows us to compensate for the fact that a deterministic exploration policy will only gather data for one action in any given state. To motivate Lipschitz regularity further, in Appendix C, we provide an example of a contextual bandit problem where the Lipschitz assumption proves crucial to identify the optimal policy. Moreover, in this section we will show theoretically that, for contextual bandit problems and under reasonable technical assumptions, the Lipschitz assumption allows us to obtain useful policy gradients in the region near the manifold covered by the deterministic exploration policy, addressing the problem of critic non-identifiability. We begin by introducing the definition of the data manifold.

$$\mathcal{M} \overset{\text{def}}{=} \{(s, \pi_E(s)) : s \in \mathcal{S}\}$$

Here, $\pi_E$ is the deterministic exploration policy. We first introduce a Lemma showing that, as the amount of training data approaches infinity, the critic function $\hat{Q}$ restricted to $\mathcal{M}$ approaches the true $Q$-function. In the Lemma below, we assume that the critic belongs to the function class $\mathcal{H}$, which contains Lipschitz functions that satisfy technical regularity conditions (see appendix B for a definition). In this section, we adopt the notation that the critic $\hat{Q}_N$ was trained on a dataset of size $N$.

**Lemma 1.** *Assume that both the true $Q$-function and the critic $\hat{Q}_N$ are in function class $\mathcal{H}$ and that the MSE critic training error is zero in the sense that $\sum_{i=1}^{N} \left( Q(s_i, \pi_E(s_i)) - \hat{Q}_N(s_i, \pi_E(s_i)) \right)^2 = 0$ for all states $s_i$ in the dataset. Assume that the distribution of training data satisfies $p(s) > 0$ for all states $s \in \mathcal{S}$. Then, with probability one, for all $(s,a) \in \mathcal{M}$, we have*

$$\lim_{N \to \infty} |Q(s,a) - \hat{Q}_N(s,a)| = 0.$$

The proof of the Lemma uses standard Rademacher tools for Lipschitz functions (von Luxburg & Bousquet, 2004; Mohri et al., 2018) and is given in appendix B.

While the result above certifies the quality of function fit on the manifold itself, in order to reason about the quality of the policy gradient, we need to have a result that can be extended to nearby points. We first define the neighborhood.

$$\mathcal{M}_\eta \stackrel{\text{def}}{=} \left\{ (s, a) : \exists (s', a') \in \mathcal{M}. \ \left\| \begin{bmatrix} s' \\ a' \end{bmatrix} - \begin{bmatrix} s \\ a \end{bmatrix} \right\| \leq \eta \right\}$$

Herre, we use the notation $\begin{bmatrix} s \\ a \end{bmatrix}$ to denote a concatenation of vectors $s$ and $a$. We now introduce a Lemma which talks about the quality of fit in the neighborhood.

**Lemma 2.** *Under assumptions of Lemma 1, for state-action pairs $(s, a) \in \mathcal{M}_\eta$, with probability one we have*

$$\lim_{N \to \infty} |Q(s, a) - \hat{Q}_N(s, a)| \leq 2\eta L.$$

The proof of the Lemma can be found in appendix B. Lemma 2 quantifies the quality of fit near the data manifold. In order to have a fit guarantee that covers the entire state-action space, we now introduce a coverage assumption on the exploration policy.

**Definition 2.** A policy $\pi_{\mathrm{E}}(s)$ achieves $\eta$-coverage if $\mathcal{M}_\eta \supseteq \mathcal{S} \times \mathcal{A}$.

For example, consider the case when $\mathcal{S} = [0, 1]$ and $\mathcal{A} = [-1, 1]$. Consider the policy $\pi_{\mathrm{E}}(s) = \sin\left(\frac{2\pi s}{p}\right)$. If we set $\eta = p$, we indeed have that $\pi_{\mathrm{E}}$ has $\eta$-coverage. Moreover, we can construct policies with $\eta$-coverage for arbitrarily small $\eta$ by setting the period of the exploration policy to be equally small.

We now introduce the following corollary, which follows immediately from Lemma 2 and the definition of $\eta$-coverage.

**Corollary 1.** *For an exploration policy which achieves $\eta$-coverage, we have*

$$\lim_{N \to \infty} |Q(s, a) - \hat{Q}_N(s, a)| \leq 2\eta L$$

*for every state-action pair in $\mathcal{S} \times \mathcal{A}$.*

The Corollary implies that, for the class of exploration policies that achieve $\eta$-coverage for $\eta$ arbitrarily small, using the algorithm as defined in equation 6, a contextual bandit problem class with a Lipschitz ground truth reward is indeed identifiable on $\mathcal{S} \times \mathcal{A}$ as per Definition 1 because we can choose an exploration policy such that $\eta$ becomes arbitrarily small. This requirement is natural in the sense that, lacking stochastic actions, we need another mechanism to ensure that the dataset tells us enough about the true $Q$ function. In practical settings, we are unlikely to have an exploration policy which achieves $\eta$-coverage, since it implies covering the whole state-action space. However, the results in this section are straightforward to extend to a setting where the coverage is restricted to a region $\mathcal{R} \subset \mathcal{S} \times \mathcal{A}$.

Finally, we proceed to quantify the error made estimating the gradients. Unfortunately, Lipschitz continuity is not enough to ensure that closeness of functions implies the closeness of gradients. To have that property, we need additional assumptions. Specifically, we choose to assume that both $Q$ and $\hat{Q}_N$ are band-limited functions (see appendix B for an exact definition). We stress that this assumption is true of many physical systems since the presence of very high frequencies in $Q$ indicates a lack of stability.

**Proposition 1.** *Assume that the exploration policy achieves $\eta$-coverage, that both the true $Q$-function and the critic $\hat{Q}_N$ are in function class $\mathcal{H}$ and that the MSE critic training error is zero. Assume that the distribution of training data satisfies $p(s) > 0$ for all states $s \in \mathcal{S}$ and that $Q$ and $\hat{Q}_N$ can be extended to $W$-bandlimited functions. For state-action pairs $(s, a) \in \mathcal{S} \times \mathcal{A}$, we have*

$$\lim_{N \to \infty} \|\nabla_a Q(s, a) - \nabla_a \hat{Q}_N(s, a)\|_1 \leq 8\pi W \eta L d_{\mathcal{A}},$$

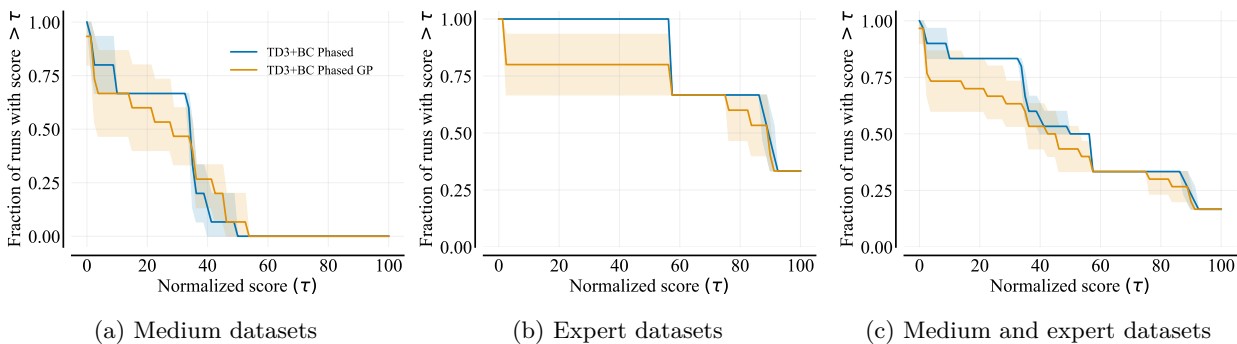

(a) Medium datasets         (b) Expert datasets         (c) Medium and expert datasets

Figure 5: Comparison of TD3+BC-Phased with and without gradient penalty (GP). Performance profiles were computes using 5 seeds in each of 3 environments (MuJoCo Walker, Hopper and HalfCheetah).

*where we denoted the dimensionality of the action space with $d_{\mathcal{A}}$.*

The proof is found in appendix B. Proposition 1 is important because it quantifies the amount of error in our estimates of policy gradient, which is what the TD3 algorithm is based on.

In the next section, we proceed to examine the effects of imposing such Lipschitz regularization in practice.

## 8 Assessing the Practical Effectiveness of Lipschitz Regularization

In the past section, we addressed the problem of critic identifiability theoretically, identifying assumptions under which we can guarantee recovery of accurate policy gradients even if the exploration policy is deterministic. In practice, the crucial assumption enabling us to claim generalization over the action space was Lipschitz continuity. In this section, we attempt to draw empirical insights from this theoretical argument. Specifically, we investigate the effect of Lipschitz regularization on the performance of the phased version of the TD3+BC algorithm.

In order to make our critic smoother, we add a gradient penalty term to the critic loss. This is inspired by the literature on the Wasserstein GAN (Gulrajani et al., 2017). In practice, we add the term $\beta \left( \frac{1}{d_{\mathcal{S}}} \|\nabla_s \hat{Q}(s,a)\|_2 + \frac{1}{d_{\mathcal{A}}} \|\nabla_a \hat{Q}(s,a)\|_2 \right)$ to the critic loss, where $\beta$ is a small constant tuned as a hyperparameter. This serves to prevent the critic from having steep gradients. While a gradient penalty term does not strictly guarantee the critic to a Lipschitz function, it is currently a state of the art technique achieving Lipschitz regularization.

We performed an experiment to assess how adding such regularization influences the performance of the batch RL algorithm in the same setting we used in section 6 (a deterministic variant of the D4RL benchmark). Results are shown in Figure 5. The experiment shows that adding Lipschitz regularization does not affect performance in a statistically significant way on the medium datasets, while causing a slight performance degradation on the expert datasets. The overarching conclusion is that batch RL practitioners now have a choice. Using TD3+BC-Phased, achieves good empirical performance, but is potentially susceptible to the critic identifiability issue. On the other hand, using the version with Lipschitz regularization means we will not be susceptible[3] to critic identifiability issues, but involves paying a price in terms of performance.

## 9 Conclusion

We have identified the critic identifiability problem, which arises when batch RL technology meant for data coming from stochastic policies is used with exploration policies that are deterministic. We also propose a solution based on Lipschitz regularization, which works for practical MuJoCo control problems and addresses

---

[3]In the idealized setting described in Section 7.

the critic identifiability problem while causing only a small loss of performance relative to applying the vanilla TD3+BC-Phased algorithm.

**Broader Impact Statement**

The social risks of deploying batch RL with deterministic exploration are similar to these of batch RL in general. Because our work is generic and tests on industry-standard benchmarks, it does not carry a significant risk of immediate harm. While RL can certainly be used for nefarious purposes, we believe those risks to be out-weighted by the possibilities of positive impact that arises from making existing control systems more efficient by using offline data. Our work does carry a small additional risk of over-reliance on the theoretical results, in settings where our assumptions are not met. We tried to mitigate these risks by spelling out the assumptions explicitly.

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
