# OpenReview forum: "Critic Identifiability in Offline Reinforcement Learning with a Deterministic Exploration Policy"
_TMLR — Rejected by TMLR_

### Review · Reviewer_ibzG · 2023-03-13

**Summary Of Contributions:**

The authors consider the offline RL setting in which the dataset is generated by a deterministic policy. They introduce the problem of critic identifiability in which the critic learns to ignore the action input, since the actions in the dataset are a deterministic function of the state. They introduce TD3+BC+Phased which learns a critic with respect to the behavior policy and then updates the policy with this critic. The authors also introduce Lipschitz regularization to the critic. Experiments are run on the D4RL benchmark and compared with TD3+BC.

**Audience:**

No

**Broader Impact Concerns:**

I have no concerns and the authors address it.

**Claims And Evidence:**

No

**Requested Changes:**

The paper suffers from several critical issues. I think there are some interesting ideas and the emprical results suggest some potential.

I do not think it is worth requesting changes as the authors should consider rewriting the paper around the proposed algorithm with new motivation and analysis.

**Strengths And Weaknesses:**

Strengths:
- The critic identifiability problem is an interesting (potential) problem.
- Writing is clear outside of theoretical sections.
- Empirical results suggest some promise.

Weaknesses

- The experimental section is based on the TD3+BC repo which uses the D4RL benchmark. The D4RL benchmark is not generated by a deterministic policy, which invalidates any claims based on experimental results.
- The authors themselves show that critic identifiability is not a problem which occurs in practice. In fact, the opposite problem has been observed in prior work (extrapolation error [1]) in which the critic generalizes too strongly based on the action.
- I'm not convinced the deterministic behavior policy setting is significantly different than the stochastic behavior policy setting, when the dataset is finite, and the state space is continuous. If we assume each state has only been viewed once, then the two settings are indistinguishable.
- The proposed algorithm is not very novel. The idea of learning from a critic of the behavior policy has been proposed in [2,3].
- There are several statements suggesting that prior work assumes total state-action coverage. This goes against the focus of most work in offline deep RL, which is dealing with incomplete coverage and finite datasets (example [1] has some explicit discussion on absent data).
- Theory seems to be based on assumptions which are not stated. Def 1 implies convergence to the true value function given sufficient data. However, deep RL methods are not even guaranteed to converge at all. Theory in section 7 is based on ideas of coverage which seems surprising given the focus of the paper was meant to be when coverage is missing. Several variables are undefined.

References:
- [1] Fujimoto, Scott, et al. "Off-policy deep reinforcement learning without exploration." 2019.
- [2] Brandfonbrener, David, et al. "Offline rl without off-policy evaluation." 2021.
- [3] Gulcehre, Caglar, et al. "Regularized behavior value estimation." 2021.

---

> ### Author Response · Authors · 2023-03-13
> **Thank you for the review!**
>
> We appreciate you say our submission has some significant weaknesses. We wanted to comment on this.
> 1. Only deterministic policies were used to generate data in the experiments. Our rollout data was prepared in a way that mirrored the one used to generate the original D4RL datasets, except we only used the policy mean during rollouts. Apologies for not making this explicit in the dataset description.
> 2. While the problem of critic identifiability surprisingly does not seem to happen for MuJoCo domains as solved by variants of the TD3+BC algorithm, it can definitely happen in practice, even with contextual bandits with linear function approximation. If you feel it is useful, we can include an example along these lines in a revised version of the paper.
> 3. To see how the setting of deterministic exploration policy is different from the setting of a stochastic exploration policy consider a contextual bandit with a continuous one-dimensional state and a continuous one-dimensional action. Consider two cases: (1) a stochastic policy that samples the action uniformly and (2) a deterministic policy that always chooses action = state. Now consider what happens if we generate data using these policies. Data from the stochastic policy will cover the whole state-action space (and will cover it more densely as we add more data). Data from the deterministic policy will only appear on the line state=action, never covering the rest of the state-action space. Given a lot of exploration data, the difference will be very stark.
> 4. We agree that the algorithm is not super-novel. We think that finding that an existing algorithm works in a new setting is a valuable contribution to the community. To quote from the TMLR acceptance criteria, "novelty of the studied method is not a necessary criteria for acceptance".
> 5. Thank you for the literature pointer. You are right when you say that your reference [1] considers the case of finite datasets / action data. We will update our discussion of state-action coverage in prior works. What we wanted to say is that all prior works we know about implicitly assume that a stochastic exploration policy is supported on the whole action space (which can be thought of as a kind of coverage). This is not the same as saying that the dataset contains all possible state action pairs (which definitely won't happen in continuous problems).
> 6. We will make the assumptions in the theory section more explicit. Our results in section 7 hold for a contextual bandit which uses Lipschitz function approximation as opposed to deep RL (as you say, proving / ensuring convergence for deep RL methods is very hard). When we discuss eta-coverage in section 7, we mean a new notion of coverage, which is needed because the policy is deterministic. In other words, we focus on the question: what requirements does the deterministic exploration policy have to satisfy to yield useful policy gradients? We will proofread the paper again to find the undefined variables.
>
> If you disagree with any of the above points, we would really welcome a reply / discussion.
> More broadly, we are interested in any hints / change requests which will make the paper better.

---

> ### Author Response · Authors · 2023-04-13
> **New version of paper.**
>
> In response to the weaknesses you identified in the review, we have made the following changes to the paper.
> 1. Dataset description was clarified to indicate we only used deterministic policies in the experiments.
> 2. We have given an example of a linear contextual bandit where the critic identifiability problem happens organically.
> 3. We have added a new figure (new Figure 1) and an explanation why the setting of deterministic exploration is different from stochastic exploration for continuous state-action spaces.
> 4. We have adjusted our claims about novelty. While the algorithm itself may not be novel, the idea of applying it to data generated by deterministic exploration policies is.
> 5. The related work section has been reworked to include research related to concentrability and other theoretical batch RL work. We do not claim prior work assumes that the dataset contains every state-action pair, but almost all of it assumes that the dataset was generated by a stochastic policy.
> 6. We have now reworked Definition 1 to make it rigorous. We do not provide theoretical results for the deep RL setting but for the more theoretically tractable setting of the Lipschitz function class, which is now clearly stated. Our notion of $\eta$-coverage is very different from the standard notion of coverage one has with stochastic exploration.
>
> We would very much appreciate it if you have any hints about how to improve the work further.

---

### Review · Reviewer_Nysj · 2023-03-22

**Summary Of Contributions:**

The paper considers Offline RL when the dataset is generated by a deterministic policy. The authors note that when the data is generated by a deterministic policy, the Q function may not be identifiable, but that inductive bias of the Q function learning algorithm may yield good Q functions anyways. The paper then presents a phased version of TD3+BC inspired by One Step RL and runs it on D4RL. Finally, some theory motivates the idea of using Lipschitz continuity of the Q functions to allow for identifiability even for deterministic policies, and a variant of the algorithm with Lipschitz regularization is presented.


**Audience:**

Yes

**Broader Impact Concerns:**

No concerns.

**Claims And Evidence:**

No

**Requested Changes:**

1. The clarity issues raised in weaknesses 1 and 2 above should be addressed.

2. There needs to be a more careful discussion and consideration of how the experiments connect to the motivation of the paper.

3. The novelty of the approach relative to the One Step RL approach of [Brandfonbrener et al., 2021] should be made clear. This likely requires adding a baseline to Figure 3.

4. The authors should provide a toy example where the Lipschitz regularization can provide a clear benefit empirically.

All of these issues are critical except for 4, which is important, but not absolutely necessary.

**Strengths And Weaknesses:**

### Strengths

1. The paper presents some interesting toy experiments in a deterministic contextual bandit problem that show how a learned Q function has inductive bias to recover a good Q function even when a pathological Q function can fit the data just as well.

2. The theory presented for Lipschitz functions is an interesting idea to motivate how generalization can accomplish some of the same things as stochasticity of the behavior policy for offline RL.

### Weaknesses

1. Definition 1 is not rigorous. First, where is the policy $ \pi_E $ used? It seems that it is used to collect $ D_N $, but this is not clear from the writing. Second, how is $\hat Q_N$ trained on the dataset? It likely depends on the function class and the optimization procedure. If that is the case, then the identifiability as presented does not only depend on the MDP, but also on the algorithm used to learn $ \hat Q_N $. This is potentially serious issue with the framing of the paper since if the Q function class is sufficiently nice, then it may resolve the issues of identifiability even with a deterministic policy (for example, as an extreme case, if the class only contains the true Q function).

2. The algorithm is unclear. The algorithm first learn $ \hat \pi_E$ and then $ \hat Q$ and $ \pi$ by equations (5) and (3) respectively. However, neither equation seems to use $ \hat \pi_E$. What is the point of learning $ \hat \pi_E$? Again this seems like a fundamental issue to understand the point of the paper. My impression is that equation (5) is likely incorrect and should be sampling $ a' $ from $ \hat \pi_E$, but it is unclear.

3. The experiments seem to depart from he motivation for the paper to begin with. The entire purpose of the paper claimed to be about studying offline RL when the data is deterministic. However, the D4RL data that is used is generated by stochastic policies. Thus, it is not clear why the experiments provide evidence that the proposed approach is actually useful because of how it deals with determinism.

4. The novelty of the approach relative to the One Step RL approach of [Brandfonbrener et al., 2021] is not clear. In particular, the TD3+BC phased algorithm that is presented seems to only differ from the One Step RL approach by estimating $ \hat Q $ of $ \hat \pi_E $ instead of directly using SARSA to estimate $ \hat Q$ of the data generating policy. This seems to be a relatively minor difference and I don't think there is a clear argument as to why the generalization of $ \hat \pi_E $ should be beneficial over the standard generalization by $ \hat Q $ trained with SARSA. To resolve this, the authors could add a One Step RL baseline to figure 3 to see if there is any difference. As written currently, Figure 3 does not seem to present any clear difference between the performance of the

5. The Lipschitz theory is a nice idea, but as the experiments show, it actually harms performance. In this sense, the math does not seem to be motivating a useful algorithmic principle. I understand that it may not help in all cases, but if the authors want to claim that this is actually the mechanism by which the algorithm works, it would be useful to provide at least a toy example where the Lipschitz regularization can provide a clear benefit empirically.

---

> ### Author Response · Authors · 2023-03-24
> **Thank you for the review!**
>
> We wanted to comment on the weaknesses you have identified with our submission.
> 1. We agree that Definition 1 needs to be reformulated, for the reasons you say. We will do this in an updated version of the paper.
> 2. Thank you for pointing this out. You are exactly right in that we made a mistake in equation (5). In fact, a' is indeed sampled from $\hat{\pi}_E$. We will make it more clear in a revised version of the paper. We note that our source code for the implementation was correct, it was just equation (5) in the paper which had this bug.
> 3. To conduct our experiments, we actually regenerated D4RL datasets from scratch using the mean of the policy (and ignoring the standard deviation). This means that the policies that generated the data in our experiments were in fact deterministic. We wrote that we used D4RL because we still used their glue code and a (modified) version of their data generation scripts. Apologies for not making this explicit in the paper.
> 4. The reason we need to behaviourally clone the expert policy is because the D4RL dataset (as returned by env.get_dataset()) does not contain the successor actions (a'). Since these are needed for SARSA updates, we need to perform behavioral cloning. We will make it clear in a revised version of the paper that we do not claim that this process leads to performance that improves on the work of Brandfonbrener et al., (2021). However, it is required to learn from the data as exposed by the D4RL interface.
> 5. We will provide an updated version of the paper with a toy contextual bandit example where the Lipschitz assumption is crucial in obtaining a good policy.
>
> We really appreciate you made a number of clear improvement requests concerning how to make a better, updated version of the paper.
> We will update the paper addressing your points once we get the third review.

---

> ### Author Response · Authors · 2023-04-13
> **New version of paper.**
>
> In response to the weaknesses you identified in the review, we have made the following changes to the paper.
>
> 1. We have made Definition 1 rigorous and included a clear dependency on the algorithm used to learn the critic.
> 2. We have clarified the algorithm description regarding how the behaviorally cloned policy is used.
> 3. Dataset description was clarified to indicate we only used deterministic policies in the experiments.
> 4. We do not claim algorithmic novelty over the work of Brandfonbrener et al. We only claim that the algorithm also works on datasets generated by deterministic exploration policies (a setting which Brandfonbrener et al. did not study).
> 5. In a new appendix of the revised version of the paper, we provided an example where Lipschitz regularisation helps.
>
> We would very much appreciate it if you have any hints about how to improve the work further.

---

### Review · Reviewer_xeFx · 2023-03-31

**Summary Of Contributions:**

This paper delves into three essential components of reinforcement learning: critic identifiability, offline reinforcement learning, and deterministic exploration policy.

The first component, critic identifiability, refers to the ability to distinguish between the value functions of two distinct policies. This problem refers to that given adequate state coverage and a smoothness assumption on the Q-function, the critic is identifiable in the sense that, as the amount of training data approaches infinity, the learned critic becomes close to the true Q-function.

The second component, offline reinforcement learning, is an approach that involves training an agent on pre-collected data without any interaction with the environment.

The third component, deterministic exploration policy, is a technique used to improve exploration in reinforcement learning. The paper proposes the TD3+BC-Phased design, a novel approach for offline reinforcement learning, that utilizes a deterministic exploration policy.

In summary, this paper offers interesting insights into three critical components of reinforcement learning and find that "the inductive bias arising from using critics represented by ReLU networks and learned using the Adam algorithm (Kingma & Ba, 2014) is enough to provide adequate generalization and useful policy gradients".

**Audience:**

Yes

**Broader Impact Concerns:**

This work considers deploying batch RL with deterministic exploration. Since this work is generic and tests on industry-standard benchmarks, it does not carry a significant risk of immediate harm.

**Claims And Evidence:**

Yes

**Requested Changes:**

I have listed all the requested changes in the above section. Please modify them carefully.

**Strengths And Weaknesses:**

I have summarized the advantages in above. In addition to that, this paper has some issues.

1. **Dataset definition** In section 2, you mentioned the setting is the discounted setting, then why there is a finite horizon $T$? The standard definition should sum up to $\infty$, and if you want to consider the finite horizon case, then there is no need to use the discount factor and simply set $\gamma=1$.

2. The claim "Finally, $Q_{\phi_1}$ is always used in the policy learning step. Learning $Q_{\phi_2}$ has no use beyond helping to reduce the overestimation bias." is a bit unnatural. Since using $Q_{\phi_2}$ can help reduce the overestimation bias, why not using it?

3. One major issue is that I don't believe describe the critic Identifiability based solely on the contextual bandit problem is adequate. In bandit problem, the reward is only a function of context and action. In RL, Q-function is not only a function of state and action, but also policy. In this case, the definition 1of Identifiability is insufficient. Which policy policy you are taking about? If you are talking about uniform over all the policy, then this is essentially the definition of uniform convergence in OPE $\sup _{\pi \in \Pi}\left|\widehat{v}^\pi-v^\pi\right| \leq \epsilon$ that as been proposed by [1],[2]. The authors need to:  (a) extends the current definition 1 to the MDP case, (b) clearly discuss the difference between your setting and the uniform OPE [1], [2] in offline RL.

[1] Near-Optimal Provable Uniform Convergence in Offline Policy Evaluation for Reinforcement Learning, AISTATS21
[2] Optimal Uniform OPE and Model-based Offline Reinforcement Learning in Time-Homogeneous, Reward-Free and Task-Agnostic Settings, NeurIPS21

4. The so-called "Identifiability" term seems not to be a new concept in offline RL. Roughly, it requires the exploration policy to explore the large enough state action space so the estimation/critic can be accurate. This is called concentrability. Without ``good'' concentrability, there is no hope that any algorithm can learn anything useful in offline RL. Could you discuss the connection of your notion to the concentrability coefficient in [3]?

[3] Information-Theoretic Considerations in Batch Reinforcement Learning, ICML19

5. "the inductive bias arising from using critics represented by ReLU networks and learned using the Adam algorithm (Kingma & Ba, 2014) is enough to provide adequate generalization and useful policy gradients." This empirical obersvation seems very disconnected to the theoretical results considered. Also, it is so confined to the Actor-Critic framework that limits its adaptivity to the RL in general.

6. "We chose a quadratic function because all smooth functions are approximately locally quadratic." Why? Is there any reference for that? This is a very rough statement that lacks scientific rigor. Linear function is not locally quadratic. Also, what do you mean by locally quadratic?

7. The second major issue is the paper spends several paragraphs on the quadratic example, how is the example representative? This function is too easy to learn therefore I don't understand why it is useful. In addition, "Surprisingly, despite the training set being confined to the policy, the generalization ability of the ReLU network combined with the optimizer turned out to be outstanding, recovering the shape of the function even in points far away from the policy that generated the data." Why? What is the metric used? How do you define the generalization is good (comparing to what baseline algorithm)? I cannot trust the conclusion by simply looking at Figure (a) and (b) since it is so subjective. Do you have some statistical metric to verify that?

8. The third major issue is the how is the TD3+BC-Phased different from TD3+BC? In Algorithm 1, you use data to learn a $\hat{\pi}_E$, but it is never used for learning $\hat{Q}$ or $\pi$? How is the behavior cloning play any role in the Algorithm1?

9. The improvement in Figure 3 seems marginal therefore it's hard to evaluate its effectiveness. Could you plot the statistical test such as Figure 9 [4] so readers can understand whether your improvement is significant?

[4] Deep Reinforcement Learning at the Edge of the Statistical Precipice, NeurIPS21

---

> ### Author Response · Authors · 2023-04-03
> **Thank you for the review.**
>
> We wanted to address the weaknesses you being up.
> 1. We used a finite-horizon discounted formulation since this is the setting in which tasks from the D4RL benchmark are typically used.
> 2. In the TD3+BC algorithm, the second critic is used in the computation of the minimum in the learning target of the critic update (to battle overestimation bias), but is not used in the actor update. This is the way the TD3+BC algorithm was introduced in the original paper by Fujimoto & Gu ("A Minimalist Approach to Offline Reinforcement Learning"). We did not change this aspect of the algorithm, instead relying on this piece of prior work by Fujimoto & Gu. If you find it useful, please reply with a comment and we can provide an ablation that compares this approach to the one that uses both critics in the actor update.
> 3. You are right when you say that the Q function in MDPs depends on the policy. We used contextual bandits as an illustration because the problem we tackle, where deterministic exploration coupled with MSE loss on the critic can lead to arbitrarily bad policy gradients, is already present in the contextual bandit case. We will provide a revised version of the paper where our definitions are more rigorous and where we make more clear which parts of the theory apply to contextual bandits and which ones to MDPs. The main difference between what we do and the work of Yin at al. ("Near-Optimal Provable Uniform Convergence...") is that they assume that the data-gathering policy is stochastic (see their section 2.2), which we do not. Similarly, Theorem 3.1 from the work of Yin and Wang ("Optimal Uniform OPE and Model-based...") assumes a problem class with minimal marginal state-action probability under exploration greater than zero. We will reference these works in an updated version of our paper and  discuss the difference.
> 4. The main difference between our work and the work by Chen and Jiang ("Information-Theoretic Considerations in Batch Reinforcement Learning") is that they assume that the exploration policy is stochastic, while we do not. Indeed, in their Assumption 1 (finiteness of the concentrability coefficient), they divide by the probability of a state-action pair under the exploration policy. We will discuss this work in a revised version of our paper. The gist of our argument is that, in certain settings, data from the "missing" actions can be replaced by regularity assumptions on the MDP (like the Lipschitz assumption). Moreover, in a revised version of the paper, we will provide a contextual bandit example where assumption 1 by by Chen and Jiang dies not hold, but where, thanks to the Lipschitz assumption, we can still obtain a useful policy from deterministic exploration data.
> 5. We agree that the connection between theory and practice could be improved, but this applies to deep RL in general, not just to our work. Providing a complete, non-vacuous, characterisation of generalisation of ReLU networks is a research challenge orthogonal to our work. Our argument is that we can study the problem formally in a simpler setting (contextual bandits, Lipschitz function class) in the hope that the learnings will translate to the full setting (MDPs, ReLu network function class).
> 6. By locally quadratic we mean "can be approximated in a neighbourhood of a point by a second-order polynomial". We use the word "locally" we mean that the approximation is only good in a small neighbourhood. Since you ask about linear functions, we also wanted to say that a linear function can be represented by a first-order polynomial exactly (and hence also by a second-order polynomial, by setting the second-order coefficients to zero). For smooth functions, the quality of this approximation can be characterised using Taylor series.
> 7. In an expanded version of the paper, we will extend the contextual bandit analysis in two ways. First, we will provide more examples of functions (other than quadratic). Second, we will introduce a quantitative way to measure generalisation.
> 8. The policy $\hat{\pi}_E$ is used to compute the next action (action taken in state s') in the SARSA update (which is used to learn the critic). Because we made a mistake in the original version of the paper, it is not clear from the current draft. This will be fixed in the revised version of the paper.
> 9. We will provide a plot of aggregate metrics inspired by Figure 9 in the work of Agarwal et al. in a revised version of the paper.

---

> ### Author Response · Authors · 2023-04-13
> **New version of paper.**
>
> First, we wanted to clarify that we do not claim that using a deterministic exploration policy necessarily achieves better learning performance than stochastic exploration. Rather, we say that in many practical settings, data collected using deterministic exploration policies is the only data that already exists and so there is a need for batch RL methods that help us leverage it. We believe that our work represents one of the first steps in addressing batch RL in this setting.
>
> We also made a number of specific changes to the paper to address the weaknesses you bring up.
>
> Concerning weaknesses 1-2, see our response to the review.
>
> Concerning weakness 3, we have made our Definition 1 much more rigorous. We have included Yin et al. and Yin and Wang in the updated related work section.
>
> Concerning weakness 4, we have provided a discussion of the concentrability coefficient in the related work section. The gist is that most works using concentrability
> implicitly assume stochastic exploration. Also, in a new appendix, we provide a contextual bandit example with discrete states and actions, where, thanks to the Lipschitz assumption, we can learn a good optimal policy despite the fact that the exploration policy is deterministic (and the dataset only contains half the data about the system).
>
> Concerning weaknesses 5 and 6, see our response to the review.
>
> Concerning weakness 7, in an expanded version of the paper, we provide two more examples of generalisation in contextual bandits. We also provide a quantitative measure of generalisation.
>
> Concerning weakness 8, we fixed the algorithm description in the revised version of the paper.
>
> Concerning weakness 9, we included additional plots (inspired by Figure 9 in Agarwal et al.)
>
> Also, while we think "identifiability" is a good name, if you feel this is really needed, we are open to changing it to "deterministic concentrability" or other such name in another revision (please reply with a comment if you think this is the case).
>
> More generally, please post a message if you feel our paper can be improved in other ways.

---

### Review · Reviewer_zN9i · 2023-04-01

**Summary Of Contributions:**

The paper studies offline RL in cases where the offline data are collected by running a deterministic behavior policy. The paper presents insights into some of the challenges associated with this setting and proposes methods to mitigate such challenges.

**Audience:**

No

**Broader Impact Concerns:**

The paper includes a broader impact statement and it focuses on a general methodology.

**Claims And Evidence:**

No

**Requested Changes:**

- The paper needs to clearly justify why it aims to study this problem. As it stands, the motivation behind the paper is unclear and the premise that existing methods do not handle deterministic behavior policies seems to be incorrect. The motivation behind the paper becomes much clearer if the authors review prior work in detail and point out the limitations that exist in those works. Perhaps the discussion of the paper (including in the abstract) should be limited to deterministic policy gradient algorithms and the goal should be proposing practical and simple algorithms useful for offline RL with deterministic policies.
- The paper needs to be written much more rigorously. For instance, Definition 1 seems inexact and incomplete. What is Q? Is it any function of s and a? Or should it be "there exists a Q-function" ....? What does $\hat Q_N$ "denotes a critic trained on a dataset" mean? Usually, it should mean $\hat Q_N$ is a function of training data. But this is not specified.

**Strengths And Weaknesses:**

**Strengths**
- The paper may reveal some new practical insights and intuition about the difficulties associated with having the behavior policy to be deterministic as well as methods that may be useful in mitigating these difficulties.
- The proposed algorithm and methods are simple.

**Weaknesses**
- The paper appears to claim that the existing offline RL works are limited to cases where the behavior policy is stochastic and exploratory. Indeed, the majority of existing offline RL papers (even those with theoretical finite-sample guarantees) *pose no restrictions* for the behavior policy to be stochastic and they can be deterministic. For instance, the offline RL problem studied in [1] even has the setting restricted to deterministic policies. This is also the case for actor-critic algorithms, which is the focus of this paper. For instance, [2] proposes an actor-critic offline RL algorithm with general function approximation and theoretical guarantees that do not require behavior policy to be stochastic.

- The paper is weak on rigor in discussion and theory. For instance, it is stated that the critic identifiability for CB can be extended to RL but extending Definition 1 to RL where Q is a function of policy is unclear. See some additional comments in the section below.

- The identifiability definition is closely connected to the uniform coverage offline RL setting, for which pessimism is no longer needed (e.g. addition of the BC term is not needed). See [1] for a detailed discussion on uniform all-policy concentrability vs. single-policy concentrability assumptions on offline data. Thus, it is unclear to me why it makes sense to study Q-function identifiability in the context of realistic offline RL settings where pessimism is needed.

- The paper is focused on offline RL with a deterministic behavior policy but the experiments are conducted on D4RL, which uses stochastic behavior policies.

- The paper does not discuss prior work in detail and misses many offline RL papers. No discussion is presented on offline RL theory papers, which seems relevant to the theory sections of the paper.

[1] Rashidinejad, P., Zhu, B., Ma, C., Jiao, J., & Russell, S. (2021). Bridging offline reinforcement learning and imitation learning: A tale of pessimism. Advances in Neural Information Processing Systems, 34, 11702-11716.

[2] Cheng, C. A., Xie, T., Jiang, N., & Agarwal, A. (2022, June). Adversarially trained actor critic for offline reinforcement learning. In International Conference on Machine Learning (pp. 3852-3878). PMLR.

---

> ### Author Response · Authors · 2023-04-03
> **Thank you for the review.**
>
> We wanted to address the perceived weaknesses you bring up.
>
> 1. The work of Rashidinejad et al. ("Bridging offline reinforcement learning and imitation learning") does implicitly assume that the exploration policy is stochastic. This is because they use the distribution shift ratio $C^\pi$ (see their definition 1), which divides by the probability of seeing a state-action pair under exploration. That ratio becomes infinite (and their results become vacuous) as the exploration policy becomes more deterministic. The work of Cheng et al. ("Adversarially trained actor critic for offline reinforcement learning."), uses (see their section 4.1.2) a concentrability coefficient $\mathcal{C}$, which uses a norm weighted by the state-action visitation probability under exploration in the denominator. This means that $\mathcal{C}$ becomes very large or diverges altogether for a tabular function class where the measure $\nu$ puts non-zero probability on an action not represented in $\mu$. This work does, however, seem to provide non-vacuous results for deterministic exploration policies under the Lipschitz function class. We will include the work in our related work section in a reworked version of our paper (we weren't aware of this paper before).
> 2. We use contextual bandits to illustrate our theory because the problem we identified, where deterministic exploration coupled with MSE loss on the critic can lead to arbitrarily bad policy gradients, is already present in the contextual bandit case. We agree we should be more precise in delineating which parts of the paper apply to contextual bandits and which ones to MDPs. We will provide an extended, more rigorous version of the paper.
> 3. We agree that the fact that pessimism / BC term needs to be justified more is valid criticism. In our experience, we can do away with pessimism / BC term is small examples where we can achieve the kind of full coverage our definition 1 is trying to model. On the other hand, on more realistic MuJoCo tasks, we do need pessimism / the BC term to obtain a useful policy. We believe that the notion of identifiability can still be useful for the more complicated tasks because, if we make the right assumptions (like the Lipschitz assumption on the Q function), even though we won't have identifiability on the full action space, we can still have identifiability on a subset of the state-action space significantly larger than the support of the exploration policy. We agree the current version of the paper does not include a discussion of this (nor do we discuss identifiability on a subset of the state-action space). We will provide a revised version that does.
> 4. The exploration policies used in our experiments were in fact deterministic (only the policy mean was used to generate rollouts). We said we used D4RL only because we used their glue code and a modified version of their data generation scripts. We agree that it wasn't clear in the original paper and will fix the dataset description in a revised version.
> 5. To the best of our knowledge, most of batch RL work explicitly of implicitly assumes a stochastic data generation policy. If you have more examples of works that provide a non-vacuous analysis of deterministic exploration in batch RL, we would appreciate if you could post pointers as a comment.
>
> Concerning your requested changes, we will provide an extensively updated version of the paper, which will (among other changes) include an update to the discussion of related work. We will also re-do the theoretical sections to improve the level of rigour.
>
> We also wanted to point out that our main motivation was the fact that in many settings in industry, datasets already exist that have been gathered by a deterministic policy and, ideally, for practical engineering reasons, we want to do batch RL on them using algorithms similar to what already exists and has been proved to work on stochastic datasets.

---

> ### Author Response · Authors · 2023-04-13
> **New version of paper.**
>
> First, we want to clarify the motivation for our work. We believe that batch RL with deterministic exploration is an understudied problem that has not been explored well in existing literature. For example, Rashidinejad et al. implicitly assume that the exploration policy is stochastic. The framework of Cheng et al. does allow for deterministic polices under the Lipschitz function class, but the paper has a different objective to our work: while we are looking for the "shortest-path" way of applying existing algorithms (built with stochastic exploration in mind) in the setting of deterministic exploration (which involves learning an accurate critic, hence the study of identifiability), they are interested in making a novel algorithm based on a notion of pessimism. Ultimately, our approach is more engineering-friendly (because one can use already existing high-performance implementations), while theirs has a more extensive theoretical foundation. We believe there is room for both in the community.
>
> Also, concerning our background section, we think we are right in discussing TD3+BC rather than just the "generic" deterministic policy gradient because TD3+BC is an algorithm that specifically targets batch RL. We also wanted to point out that the word "deterministic" in deterministic policy gradient does not necessarily imply a deterministic data-gathering policy, only the fact that the policy the algorithm is learning about is deterministic.
>
> We have now uploaded a revised version of the paper, which is written in much more rigorous way.
>
> We have also made a number of specific changes to the paper to address the weaknesses you bring up.
> 1. We have updated the related work section in the paper to include Rashidinejad et al., Cheng et al. (as well as other works).
> 2. We have redone Definition 1 to be more rigorous.
> 3. The study of identifiability is needed because we want to use algorithm with existing high-quality implementations (like TD3+BC) which rely on a good critic to work. We have extended the notion of identifiability to a subset of the state-action space in order to make it closer to what is required for realistic tasks.
> 4. We have clarified the dataset description to indicate that we used deterministic policies to gather data.
> 5. We have expanded the related work section to include theory papers.
>
> We would appreciate if you have further suggestions on how to improve our work.

---

### Author Response · Authors · 2023-04-13
**Thanks again for the constructive reviews and allowing us to resubmit an updated version of the manuscript.**

We have now uploaded a new, revised version of the paper. Highlights are listed below.
1. A rewritten definition 1, which now has the required level of rigour.
2. A reworked related work section that now incudes more theoretical work.
3. More examples of contextual bandit reward functions.
4. An en example where the critic identifiability problem emerges organically when the critic is learned with gradient descent.
5. We clarified how the behaviorally cloned policy is used in the TD3-BC-Phased algorithm.
6. A new appendix which shows a bandit example where the Lipschitz assumption is crucial for learning the optimal policy,
7. An expanded Figure 4, which shows Median, Mean and IQM measures of performance, together with confidence bars.

We also replied to each reviewer separately, outlining how we addressed their concerns.

---

### Decision · Action_Editors · 2023-05-28

**Recommendation:** Reject

**Comment:**

**Summary:**

The paper studies offline (batch) RL under a deterministic exploration (behaviour) policy. A deterministic policy does not provide adequate coverage of the action space. The paper defines a notion called Identifiability and argues that it is not satisfied under a deterministic policy. It empirically shows that despite lack of coverage, a neural network can learn an accurate function on some toy contextual bandit examples.

It then suggests two algorithms.
One is TD3+BC+Phased, a simplified variant of the TD3+BC algorithm, with the rationale that if the exploration policy has been deterministic, the critic should not be relied upon beyond the data distribution of the exploration policy. This algorithm then performs one iteration of policy improvement.

The second algorithmic suggestion is based on the argument that if the underlying Q functions are Lipschitz, one can generalize beyond the data manifold and still have a controlled amount of error in approximation. This then leads to an algorithm with a regularized critic that (indirectly) controls the Lipschitzness of the estimated Q function. The results do not show improvement compared to when there is no gradient penalty.



**Evaluation:**

The reviewers, who are all experts in the area from both theoretical and empirical perspectives, have raised many issues with the paper. Consequently, the authors have significantly revised the paper. The reviewers' final recommendations, however, are still mixed: there are one Accept, one Leaning Accept, and two Leaning Rejects.


After going through the reviews and the revised paper, I can see that some of the issues have been addressed satisfactorily. For example,
- The confusion about D4RL benchmark being based on stochastic policy is resolved.
- Some reviewers mentioned that the quadratic example is too simple. The authors have added some new more complicated examples.
- Some typos in the definition of algorithm have mostly been resolved.
- There have been some concerns about the novelty of the algorithm. The authors have clarified this. This not the main criteria for acceptance after all.


But there are still some remaining issues in the paper. For example,

- The reviewers find the motivation behind the work and algorithms unclear, even after the revisions.
- They do not find the algorithm well motivated to address the proposed problem.
- The theoretical definitions and results are not always rigorous.
- The claim that the work is the first paper that studies batch/offline RL in a setting where the exploration policy is deterministic is inaccurate. The authors have added some new papers in their revised version, but it is still not satisfactory to some of the reviewers.
- The empirical evaluations is limited, the benchmark is limited to only a single baseline, and the proposed algorithm(s) do not substantially outperform baselines on benchmark RL problems beyond the carefully constructed toy bandit problems.



I realize that the authors have real-world motivation behind their work, and I don't think that is the sense reviewers feel confused about the motivation. I believe they are confused more about how that motivation has led to specific definitions and algorithms.
Also I realize that the new version has revised theoretical statements, which are more accurate, though I believe there are still several inaccuracies in the mathematical definitions, statements, and proofs, as I will explain some of them later.
I also acknowledge the authors' private comment on OpenReview that their work has "useful nugget of knowledge". I read the paper and I agree with it. I believe there are valuable insights in this work.

The problem, in my opinion, is that the paper has not been written well enough to communicate the intended idea clearly. When at least three out of four reviewers have issues with the motivation of this work, and I agree with them, it suggests that the authors need to go back to the drawing board and restructure and rewrite their paper so that their ideas are better communicated.

Overall, I believe that even though the paper has some interesting lessons and some of its insights might be appreciated by the community, it is not yet ready to be published at TMLR. I encourage the authors to consider the reviewers' comments, and try to figure out what made them confused or less than enthusiastic about the current version of the paper, and then try to present their work to address them. **If they do a major revision, I encourage them to resubmit.**



**Additional Comments:**

In addition to the comments by the reviewers, I have some further comments about the work, which may help the authors improve their work:


Some comments about the notation of Identifiability:

- This definition is some version of the the pointwise (or supremum norm) consistency of the estimator. It is good to acknowledge it as such.
- Definition 1 seems to have a mistake in the order of quantifiers. It states that "... for every $\epsilon > 0$ ... there exists an exploration policy and a dataset ..." such that the error between the estimate and the true value of the policy is smaller than $\epsilon$.
This means that we can choose the exploration policy ourselves and then require to have a good estimate of that particular policy. But this is not what is needed in batch/offline RL. The exploration/behaviour policy is fixed, and the goal is to estimate the current policy, produced by the actor, with good precision.
- Beside this issue, this definition is a strong requirement, and it is not clear whether it is the right characterization of the problem. Let me explain this more:

If we have Identifiability, we can estimate the true $Q^\pi$ with arbitrary precision over all state-action pairs. That is, we know the true value function for any policy. If that is the case, guaranteeing the convergence of the policy to the optimal policy is easy. For example, we can run the exact Value/Policy Iteration under this assumption, which is guaranteed to converge with a certain rate.
In other words, "Identifiability" implies "Optimal policy can be found".
On the other hand, Non-Identifiability does not imply that Optimal Policy cannot be found.
Some critics may very well not satisfy this definition of Identifiability, and yet be enough to learn a good policy. For example, this may be the case when the critic is accurate enough according to an $L_2(\mu)$ norm, instead of $L_\infty$, as required in this definition, if the distribution $\mu$ is where the current policy of actor happens to go.

This suggests that Identifiability may not be the right way to characterize what makes an batch RL algorithm work or break.

- A related issue is that how this paper uses this definition after all. There is no formal statement requiring Identifiability. We have a mathematical definition without any formal statement built on top of it. What purpose does it serve then?
(I should say that since Identifiability is a very strong condition, one can show that if it holds, we can solve the MDP using Value Iteration/Policy Iteration up to arbitrary precision, which is not a surprising result).



Some comments about Lemma 1:
- The assumptions that MSE critic error is zero is strong, for many reasons, including that as opposed to the usual regression problem, where the target is directly accessible, we do not have that in the RL setting using a bootstrapped estimate (TD).  The reason the analysis of variants of the Fitted Q-Iteration algorithm is not simply the same as the analysis of a regression problem in the supervised learning setting is exactly because of this reason.
An exception is a Monte Carlo estimate of the value function, which reduces the critic to a regression estimator.

- The statement has both summation being zero and "for all states $s_i$ in the dataset".

- The last paragraph of the proof of Lemma 1 should have used the Borel-Cantelli lemma to be rigorous.


Some comments about Proposition 1:
- This shows that the gradient of $\hat{Q}$ w.r.t. action is close to the gradient of $Q$ w.r.t. actions. What we care, however, is the closeness of the policy gradients. Some applications of the chain rule are needed.

- The meaning of "can be extended" in the statement of the proposition should be clarified. I noticed that it is in the supplementary material, but at least an intuition should be provided.




Some comments about Appendix C:

It is shown that on a specially designed bandit problem, Lipschitz continuity helps. The example is not a good one because even though the state/action space is supposed to be continuous when we talk about Lipschitzness, this contextual bandit problem has a finite state and action spaces. Those spaces are embedded in the Euclidean space, so one can still talk about continuity, but this is not the spirit of how Lipschitzness might help.



Lipschitzness can mitigate the lack of data coverage:
One key message of this paper is that even if we do not have a data coverage, Lipschizness can allow the estimator to gracefully generalize beyond the data manifold, and satisfy some approximate Identifiability. Given that their notion of Identifiability is very similar to pointwise consistency, I encourage the authors to take a look at the literature on pointwise consistency. Many of those results are expressed for a nonparametric estimator, but reading them might be insightful. An example is the following paper

Laszlo Gyorfi, "The Rate of Convergence of k-NN Regression Estimates and Classification Rules," 1981.

(To be clear, this is not a request to cite this paper.)

**Audience:**

Reinforcement learning (RL) in the batch (aka offline) setting is an important topic in the RL community. This paper could be of great interest to them.

**Claims And Evidence:**

Some of them, but not all. The clarity of the paper and its storyline has some room for improvement.